# Recombinant Fowlpox Virus Expressing gB Gene from Predominantly Epidemic Infectious Larygnotracheitis Virus Strain Demonstrates Better Immune Protection in SPF Chickens

**DOI:** 10.3390/vaccines8040623

**Published:** 2020-10-22

**Authors:** Sujuan Chen, Nuo Xu, Lei Ta, Shi Li, Xiang Su, Jing Xue, Yinping Du, Tao Qin, Daxin Peng

**Affiliations:** 1College of Veterinary Medicine, Yangzhou University, Yangzhou 225009, China; chensj@yzu.edu.cn (S.C.); DX120190136@yzu.edu.cn (N.X.); tl940716@163.com (L.T.); 18437958221@163.com (S.L.); MX120190726@yzu.edu.cn (X.S.); MZ120181094@yzu.edu.cn (J.X.); dyp@yzu.edu.cn (Y.D.); qintao@yzu.edu.cn (T.Q.); 2Jiangsu Co-Innovation Center for the Prevention and Control of Important Animal Infectious Disease and Zoonoses, Yangzhou University, Yangzhou 225009, China; 3Joint International Research Laboratory of Agriculture and Agri-Product Safety, the Ministry of Education of China, Yangzhou University, Yangzhou University, Yangzhou 225009, China; 4Jiangsu Research Centre of Engineering and Technology for Prevention and Control of Poultry Disease, Yangzhou University, Yangzhou 225009, China

**Keywords:** infectious laryngotracheitis virus, glycoprotein B, recombinant fowlpox virus, immune efficacy, chickens

## Abstract

*Background*: Infectious laryngotracheitis (ILT) is a highly contagious acute respiratory disease of chickens. Antigenic mutation of infectious laryngotracheitis virus (ILTV) may result in a vaccination failure in the poultry industry and thus a protective vaccine against predominant ILTV strains is highly desirable. *Methods*: The full-length glycoprotein B (gB) gene of ILTV with the two mutated synonymous sites of fowlpox virus (FPV) transcription termination signal sequence was cloned into the insertion vector p12LS, which was co-transfected with wild-type (wt) FPV into chicken embryo fibroblast (CEF) to develop a recombinant fowlpox virus-gB (rFPV-gB) candidate vaccine strain. Furthermore, its biological and immunological characteristics were evaluated. *Results*: The results indicated that gB gene was expressed correctly in the rFPV by indirect immunofluorescent assay and Western blot, and the rFPV-gB provided a 100% protection in immunized chickens against the challenge of predominant ILTV strains that were screened by pathogenicity assay when compared with the commercialized rFPV vaccine, which only provided 83.3%. *Conclusion*: rFPV-gB can be used as a potential vaccine against predominant ILTV strains.

## 1. Introduction

Infectious laryngotracheitis (ILT) is a highly contagious acute respiratory disease in chickens. Infectious laryngotracheitis virus (ILTV), classified as Gallid herpesvirus I, is a key pathogen, which has the characteristics of rapid spread, high mortality, and low egg production rates in infected laying hens [1]. Additionally, it is becoming an enzootic in China [2,3,4,5]. Nowadays, most of the vaccines against ILT are modified, live attenuated viruses derived from sequential passage in cell cultures or chicken embryos [6,7,8]. However, the live attenuated vaccines still possess potential risks to become virulent again [9,10], and some of the attenuated live ILT vaccines have shown side effects, such as spreading vaccine virus to non-immunized animals, finally leading to outbreak and epidemic of ILT [11]. Therefore, there is an urgent need to develop safer and more efficacious ILTV vaccines.

The fowlpox virus vector has unique advantages, including large capacity for foreign genes’ expression, efficient immune responses, strict host range, no risk of spreading virus to the environment, and an ability to distinguish infected and immunized animals [12,13]. Fowlpox virus has been a vector to successfully express protective immunogen genes from several avian viruses, including Marek’s disease virus [14,15], Newcastle disease virus [16,17], Avian Influenza virus [18], and ILTV [19,20,21]. Although there is a commercial recombinant fowlpox virus vector expressing gB gene in China, it can only provide partial protection against predominant ILTV strains, because the virulent wild-type ILTVs had been isolated from chickens that had been immunized with the commercial vaccine. Therefore, we chose the gB gene from predominantly epidemic strains and constructed a transfer vector expressing the gB gene of ILTV, obtained a recombinant fowlpox virus (rFPV) by transfection with wild-type fowlpox virus (wt-FPV), and evaluated its immune response against predominantly epidemic strains.

## 2. Materials and Methods

### 2.1. Ethics Statements

All bird experiments were approved by the Jiangsu Administrative Committee for Laboratory Animals and were conducted in compliance with the guidelines of laboratory animal welfare and ethics of the Jiangsu Administrative Committee for Laboratory Animals (Permission number: SYXKSU-2016-0020).

### 2.2. Plasmid, Viruses, Cells, gB Protein, and Experimental Animals

The fowlpox virus vector plasmid p12LS, which contains FPV1, FPV2, the expression cassette of P11-LacZ reporter gene, and the promoter Ps, was developed by Sun et al. [22]. A FPV vaccine strain 282E4 (wt-FPV) was purchased from China Institute of Veterinary Drug Control (Beijing, China). The infectious laryngotracheitis virus Gallid herpesvirus 1 strain I6E1 (I6E1), Gallid herpesvirus 1 strain I1E1 (I1E1), and Gallid herpesvirus 1 strain I19 (I19) were isolated from the vaccinated commercial layer and identified by our laboratory in Eastern China (two strains were from Shandong province, one was from Jiangsu province, and another one was from Anhui province). The virus Gallid herpesvirus 1 strain WG (WG) was donated by Professor Yantao Wu. Fertilized White Leghorn-specific pathogen-free (SPF) eggs were purchased from Beijing Meria Vitong experimental animal technology co., LTD (Beijing, China). Chicken embryo fibroblast (CEF) cells were prepared and maintained in M199 medium (HyClone, UT, USA) with 4% fetal bovine serum (HyClone, UT, USA). The cell was cultured at 37 °C in 5 % CO_2_. The plaque-forming unit (PFU) of rFPV-gB or wt-FPV was determined by inoculation of 10-fold serial dilutions into CEF cells. The 50% chicken embryo infection dose (EID_50_) of the viruses were calculated by inoculating serial 10-fold dilutions of virus into 10-day-old SPF embryonated chicken eggs through chorioallantoic membrane (CAM). The gB protein was expressed by prokaryotic expression vector pET-32a(+) in our laboratory (data not shown). 28-day-old White Leghorn SPF chickens were purchased from Zhejiang Lihua Agricultural Science and Technology Co. LTD (Ningbo, Zhejiang, China). All chickens were housed in our school’s animal facility under a standard animal study protocol.

### 2.3. Sequence Analysis of I6E1 and WG Strain

Class 1–10 sequences of ILTV were downloaded from National Center for Biotechnology Information (NCBI) [10]. The thymidine kinase (TK), open reading frame (ORF) B-TK (ORFB-TK), Infected cell protein 4 (ICP4), and gB genes and their sequences from WG strain or I6E1 strain were obtained by polymerase chain reaction (PCR) amplification and Sanger sequencing. All sequence homology was analyzed by DNAstar.

### 2.4. Construction of rFPV-gB

The full-length gB gene was amplified from ILTV by PCR using primers 5’-AAAGGATCC**GCCACC**ATGGCTAGCTTGAAAATAAA-3’ (*Bam*HI is indicated by underline and Kozak sequence is indicated by bold) and 5′-GCCCTCGAG**ATAAAA**TTATTCGTCTTCGCTTTCT-3’ (*Xho*I is indicated by underline and early transcription termination signal of FPV is indicated by bold). The resulting PCR product was cloned into pEASY-Blunt3 vector (TransGen Biotech, Beijing, China) to obtain plasmid T3-gB0.

The two sites (741–747 and 2106–2112) of FPV transcription termination signal sequence on the gB0 gene were mutated synonymously with the mutation kit using the primers 5’-GCGACAGGTGATACAGTAGAAATTTCTCCTTTCTATACC-3’ (TTTTTAT-TTTCTA) (Mutated nucleotide site is indicated by underline), 5’-TGGTCCGGTCGTGTTTTTGGTATAGAAAGGAGAAATT-3’, 5’-GCAATCTTTAGAGCAATAGCAGATTTCTTTGGCAAC-3’ (TTTTTTT-TTTCTTT) (Mutated nucleotide site is indicated by underline), and 5’-TACTTCTTCAAGAGTGTTGCCAAAGAAATCTGCTATTG-3’ respectively, and their sequences were confirmed by sequence analysis to get the plasmid T3-gB.

After digestion with *Bam*HI and *Xho*I, the mutated synonymous gB gene of ILTV was inserted into the plasmid, p12LS, which had been digested with the same restriction enzyme, to form the transferring vector, p12LSgB. The positive plasmid was confirmed by sequence analysis. Then, the vector was used to transfect into chicken embryo fibroblast (CEF) by liposome, which was pre-infected with wt-FPV. The purified rFPV-gB was obtained by blue-white plague selection.

### 2.5. Characterization of rFPV-gB

The DNA of recombinant fowlpox virus was extracted and subjected to PCR for confirming the insertion of the gB gene in the chromosomal DNA of FPV. Expression of ILTV gB was confirmed by indirect immunofluorescence assay (IFA) and Western blot using a chicken polyclonal antibody against the gB gene of ILTV.

To detect the growth kinetics of FPVs, CEF cells were infected with wt-FPV and rFPV-gB at a multiple of infection (MOI) of 0.02. After 1 h of fowlpox virus adsorption, cells were washed twice with phosphate-buffered saline (PBS) to remove unbound virus particles and the Dulbecco’s dodified eagle medium (DMEM) medium containing 1% fetal bovine serum (FBS) was added. Within the designated hours post-infection (h p.i.), aliquots of the cell supernatants were collected and the PFUs per ml were determined in CEF cells using the method of Reed and Muench [23].

### 2.6. Immunization and Challenge Study

#### 2.6.1. Challenge Efficacy of ILTVs

##### Amplification and Determination of EID_50_ of ILTVs

Four ILTV strains, I1E1, I6E1, WG, and I19, were inoculated on the 10-day-old SPF chicken embryos chorioallantoic membrane (CAM), 10 chicken embryos were inoculated with each virus, cultured at 37 °C for 120 h. The CAM containing opaque plaques were collected, ground, and freeze-thawed three times, and the supernatant was harvested to determine the EID_50_ of ILTVs.

The ILTVs were diluted for 10-fold dilution with PBS containing antibiotics, and 10^−3^, 10^−4^, 10^−5^, 10^−6^, and 10^−7^ dilutions were inoculated into 10-day-old SPF chicken embryos. Five embryos were inoculated with each dilution, and the inoculation dose was 0.2 mL/embryo. After culture at 37 °C for 120 h, the pox plaques of CAM of each embryo were observed, and the EID_50_ was calculated by the Reed–Muench method.

##### Challenge Efficacy of ILTVs in SPF Chickens

A total of 28 7-week-old SPF chickens were randomly divided into 7 groups of 4 chickens each. The chickens in groups 1–4 were inoculated by nose/eye drop with I1E1, I6E1, WG, and I19, respectively, at a dose of 10^5^ EID_50_/bird in a 0.2 mL inoculum. The chickens in groups 5 and 6 were inoculated by oropharyngeal drop with WG and I19, respectively, at a dose of 10^6^ EID_50_/bird in a 0.2 mL inoculum. The chickens in group 7 were inoculated by oropharyngeal drop with 0.2 mL of sterile PBS/bird.

Following inoculation, chickens were monitored daily and assessed for general health and clinical signs within 10 days post-inoculation (d.p.i), and all chickens were clinically scored in combination with the changes of necropsies. Scoring involved picking birds up individually for approximately 30 s to assess demeanour, respiratory signs, and any signs of conjunctivitis. The scoring criteria [9,24] were given according to respiratory signs and any signs of conjunctivitis: demeanour was scored as 0 (normal) to 2 (severely depressed), respiratory signs were scored from 0 (none) to 2 (severe respiratory distress), conjunctivitis was scored from 0 (none) to 2 (severe), and death was scored as 3. Clinical indexes (total score/total number of chickens), morbidity (number of infected chickens/total number of chickens), and mortality (number of dead chickens/total number of chickens) were calculated.

#### 2.6.2. Protective Efficacy of rFPV-gB against ILTVs in SPF Chickens

A total of 40 28-day-old SPF chickens were randomly divided into 4 groups and the chickens in groups 1 and 2 were immunized subcutaneously with rFPV-gB and wt-FPV at a titer of 10^5^ PFU in a 0.2 mL inoculum. The chickens in group 3 were inoculated on the skin of the wings with commercial ILTV recombinant fowlpox virus engineered vaccine (crFPV) at a titre of 10^5^ PFU as a positive control. Meanwhile, the chickens in group 4 were inoculated subcutaneously with 0.2 mL of sterile PBS as a negative control.

Antibodies to ILTV were detected by enzyme-linked immunosorbent assay as previously described [25]. The serum samples from immunized chickens were collected at 7, 14, and 21 days after vaccination.

At day 21 post-vaccination, each chicken was challenged with 0.2 mL of 10^5^ EID_50_ of ILTV strain I19 by oropharyngeal drop. The chickens were monitored daily for 10 days for survival and clinical signs of infection, and clinical scores were given in combination with the changes of necropsies. Furthermore, oropharyngeal swabs were collected for virus isolation from each group at 3, 5, and 7 days post-challenge. The swabs were placed in PBS and an aliquot was titrated by inoculation of embryonated eggs through CAM.

### 2.7. Statistical Analysis

The morbidity survival rates of vaccination groups were compered by Fisher’s exact test. *p*-values < 0.05 were regarded to be statistically significant, a = *p* < 0.05, b = *p* > 0.05.

Antibody titers to ILTV after vaccinations were analyzed by Graphpad 2-way analysis of variance (ANOVA), * = *p* < 0.05, **** = *p* < 0.0001.

## 3. Results

### 3.1. Sequence Analysis of I6E1 and WG Strain

The I6E1 strain has a genetic distance from the commercial vaccine WG strain, according to Agnew-Crumpton’s classification method [10], the I6E1 strain belongs to class 8 and the WG strain belongs to class 4 (Table 1).

### 3.2. Construction of rFPV of ILTV gB Gene

The positive transferring vector plasmid p12LS-gB was successfully determined by digestion with *Bam*HI and *Xho*I, including the transferring vector (10,376 bp), the target gene (2700 bp) (Figure 1), and sequence analysis (GenScript, Nanjing, China).

### 3.3. Characterization and Expression of ILTV gB Gene in rFPV

The rFPV-gB was confirmed by PCR with specific primers for the gB gene (data not shown). The rFPV-gB was screened for the presence of blue plaques, purified by seven rounds of by blue-white plague selection. The expression of the gB protein in the rFPV was determined by IFA (Figure 2). The CEF cells infected with rFPV-gB were observed by a specific fluorescence when stained with a polyclonal antibody against ILTV. By contrast, no specific fluorescence was shown in the CEF cells infected with wt-FPV, demonstrating that the gB protein was expressed correctly in CEF cells.

The cells were collected for Western blot analysis after being infected with rFPV-gB or wt-FPV, and the results showed that the CEF cells infected with rFPV-gB appeared in obviously specific bands (110 kDa) when stained with a polyclonal antibody against ILTV, but no bands were found in the cells infected with wt-FPV (Figure 3).

### 3.4. Virus Growth Curve

The PFUs of rFPV-gB and wt-FPV were determined by infection of the CEF cells. The results showed that the titer of rFPV-gB was higher than 10^6^ PFU/mL, and its yield was consistent with that of wt-FPV.

CEF cells were inoculated with rFPV-gB or wt-FPV at a MOI of 0.02. The growth curve showed that there was no significant difference between rFPV-gB and wt-FPV at each time point (Figure 4), indicating that gB gene insertion did not influence the growth property of FPV.

### 3.5. Challenge Study of ILTVs in SPF Chickens

#### 3.5.1. Clinical Signs and Throat Tracheal Lesions of Chickens after Challenge

To select the prevalent virulent strain used for challenge, the pathogenicity assay of ILTV strain was carried out in SPF chickens. The results demonstrated that part of the SPF chickens infected with I1E1 and I6E1 showed mild clinical symptoms with a slight hoarse voice or rhinorrhoea, and all the diseased chickens did not die and recovered within 10 days post-infection. After autopsy, some chickens were observed to have mucoid tracheitis, mild conjunctivitis, mild respiratory rales, and some bleeding spots on the tracheal mucosa. By contrast, the SPF chickens challenged with WG and I19 by nose/eye drop showed acute clinical symptoms, all of them presented symptoms of conjunctivitis, mainly unilateral conjunctivitis, including tears, red eyelid, swelling of orbital sinus, nasal discharge, eye secretion from serous to stick purulent, or even eyelid adhesions. While, most of them challenged with WG and I19 by oropharyngeal drop were characterized by dyspnea, mainly manifested as open-mouth-breathing, extend neck, and shake head, and some diseased chickens died because of suffocation. After autopsy, some chickens were observed to have a large amount of mucus and bleeder, caseous embolism, or blood viscous exudate lesions on the larynx and trachea (Table 2). The symptoms and lesions of the diseased chickens infected by oropharyngeal drop were observed to be more severe than those infected by nose/eye drop (Figure 5).

#### 3.5.2. The Clinical Index, Morbidity, and Mortality of I1E1, I6E1, WG, and I19

The clinical index, morbidity, and mortality of the experimental SPF chickens challenged by nose/eye drop in group WG and I19 were higher than those in group I1E1 and I6E1. Meanwhile, the pathogenicity of SPF chickens by laryngeal challenge was stronger than that of nose/eye drop in group WG and I19. Therefore, I19 was selected as the challenge strain in the subsequent challenge-protection experiment for evaluating the immune efficacy of rFPV-gB against ILTV (Figure 6).

### 3.6. Protection Induced by Immunization with rFPV-gB

#### 3.6.1. Antibody Responses to ILTV after Vaccination

The results showed that the titer of the gB antibody induced by rFPV-gB at 7 and 14 days post-infection (d.p.i) was significantly higher than that of crFPV, and there was no difference between them at 21 d.p.i (Figure 7).

#### 3.6.2. Clinical Signs and Laryngotracheal Lesions of Chickens after Immunization with rFPV-gB against ILTV Strain I19

The SPF chickens were challenged with I19 strain on 21 d.p.i, and observation was persistently carried out twice daily from day 1 to 10. The results showed that 4 out of 10 chickens in the commercial vaccine group were diseased. Three diseased chickens showed clinical signs including bad spirit, watery eyes, mild conjunctivitis, mild swelling of infraorbital sinuses, mild mucoid tracheitis, persistent nasal discharge, and respiratory rales, and one of them presented an acute respiratory symptom and died on 4 d.p.i. Three out of ten chickens in the rFPV-gB group developed mild symptoms and no death. However, all the chickens in the wt-FPV and PBS groups were infected, presenting acute respiratory symptoms. On the third day after challenge, the chickens began to present with respiratory rales, asthma, and other symptoms. From the fourth to the fifth day, the infected chickens were observed to present clinical symptoms of severe dyspnea, nasal discharge, moderate-to-severe hemorrhagic conjunctivitis, blood-stained mucus, and death. From the eighth to the tenth day after inoculation, some of the diseased chickens began to gradually recover.

Gross lesions of the dead chickens were found in the larynx and trachea, including hemorrhage in the tracheal lumen or mucoid casts. Some of them extended the whole length of the trachea, with no obvious pathological changes in other organs. By contrast, gross lesions of the rest of the vaccinated chickens consisted of swelling, hyperemia, and some small bleeding points on the laryngeal and tracheal mucosa, a little mucus in the tracheal lumen, and no pathological changes in other organs.

#### 3.6.3. The Clinical Index, Morbidity, and Mortality of Chickens after Immunization with rFPV-gB against ILTV Strain I19

The clinical indexes of rFPV-gB, commercial rFPV vaccine, wt-FPV, and PBS were 0.6, 0.3, 2.5, and 2.4, respectively. The incidences of them were 40%, 30%, 100%, and 100% respectively, and the mortality rates of them were 10%, 0%, 60%, and 60%, respectively (Table 3).

#### 3.6.4. Protection Percentage of SPF Chickens by Immunization with rFPV-gB

The protection index of SPF chickens inoculated with rFPV-gB was 100%, higher than that of SPF chickens inoculated with the crFPV engineered vaccine (83.3%). By contrast, SPF chickens inoculated with wt-FPV or PBS were all sick and died, within 14 days post-challenge (Table 4).

#### 3.6.5. Virus Shedding

Virus shedding was detected from oropharyngeal swabs in SPF chickens that survived for at least 7 days post challenge. The SPF chickens immunized with rFPV-gB showed a reduced percentage of virus shedding in the oropharynx compared to those immunized with the crFPV engineered vaccine on the third day post-challenge, and a similar percentage of virus shedding when compared on the fifth or seventh day post-challenge. By contrast, the challenge virus was able to be isolated from oropharyngeal swabs of all the SPF chickens inoculated with wt-FPV and PBS (Table 5).

## 4. Discussion

The genetic distance had been found between predominant wild ILTV strains isolated from the vaccinated commercial layer and the vaccine strain WG in China. According to Agnew-Crumpton’s and Choi’s classification method [10,26,27], the results showed that predominant wild ILTV strains were close to class 8 by phylogenetic analysis of TK, ICP4, ORFB-TK, and the gB gene, while WG strain belonged to class 4 (Table 1), which indicated that the update of the vaccine was necessary because the ILTV virus was constantly evolving.

Two FPV transcription termination signal sequences, TTTTTNT [28], in the selected full-length ILTV gB gene were found by sequence analysis, and two synonymous mutations at these two sites were performed using the point mutation kit to ensure the normal expression of the gB gene in FPV vector. Finally, the successful expression of gB protein was detected by the results of IFA and Western blot. Thus, the alteration of the TTTTTNT sequences without changing the encoded amino acids resulted in a production of full-length early mRNAs for improving the protein expression and a more consistent immune response [29].

The growth curve showed no difference between rFPV-gB and wt-FPV, indicating that the insertion of the ILTV gB gene did not affect the replication of the recombinant fowlpox virus, which is consistent with the study of other researchers [30,31,32]. When choosing animal models for challenge experiments, ILTV wild strains were mainly screened through phylogenetic analysis of the gB gene and evaluation of pathogenicity of various strains in SPF chickens, and the newly isolated strain I19 with strong virulence and typical symptoms on the SPF chickens was chosen as a challenge virus. Our study also showed that chickens exposed via trachea or ocular inoculation routes with the same ILTV and the same dose were found to have different morbidity, mortality, and virus shedding rates [33]. Therefore, the trachea exposure was chosen as the infection route to evaluate the immune efficacy of rFPV-gB in SPF chickens.

To evaluate the immunity induced by rFPV-gB, the clinical scores of chickens by observing and recording the clinical symptoms, death, and necropsy differences after challenge, combining with the virus shedding rate, were determined. The results of the clinical scores make an important contribution towards a better way of determining the pathogenicity of ILTV in chickens that may help to improve the diagnosis and control of this disease [9]. The results showed that the protection index of SPF chickens inoculated with rFPV-gB was 100%, higher than that of SPF chickens inoculated with the crFPV engineered vaccine (83.3%). This may be due to the genetic distance between the I6E1 strain and the commercial vaccine WG strain and the presence of two early termination signals of the fowlpox virus in the gB gene of commercial vaccines, which was found by sequencing. Thus, the rFPV-gB vaccine mitigated clinical signs more effectively when administered subcutaneously than the rFPV engineered vaccine [14].

The results showed that both the rFPV-gB and crFPV engineered vaccines were not more efficient in reducing virus shedding, which is consistent with the previous study [14]. Therefore, the likelihood of ILTV being spread mechanically during catching and transportation is increased if the vaccines did not prevent viral replication in the trachea and shedding into the environment. The possible strategy to improve the efficacy of current commercial ILTV viral vector vaccines is to co-administer or co-express immunomodulators that may enhance both humoral and cell-mediated responses and consequently may elicit better protection. Interleukin and interferon were often used as molecular adjuvants to enhance immune efficacy [34]. For example, an FPV vector co-expressing H5 AIV HA protein and IL-6 administered significantly enhanced the protective efficiency against AIV challenge [35]. A recombinant FPV co-expressing type I interferon and the Newcastle disease virus HN and F proteins increased the protective efficacy of the vaccine against Newcastle disease virus challenge [36]. Another study showed a rFPV co-expressing S1 glycoprotein of infectious bronchitis virus and chicken IL-18, which can significantly enhance cellular immune response induced by the rFPV-S1 [37]. In mammals, for a recombinant fowlpox virus co-expressing P12A and 3C of FMDV and swine IL-18, the immune efficacy was also increased by the expression of IL-18 [38]. Interferon has a similar effect, and Wang’s study demonstrated that chicken type II interferon increased the immune protective efficacy of a recombinant fowlpox virus co-expressing IBV-S1 and chicken IFNgamma (rFPV-IBVS1-ChIFNγ) and normal weight gain in vaccinated chickens [39]. In addition, vaccination with CpG-adjuvanted avian influenza virosomes promotes antiviral immune responses and reduces virus shedding in chickens [40]. VP6, the protein that comprises the intermediate capsid layer of the rotavirus particle, together with attenuated Escherichia coli (*E. coli)* heat-labile toxin LT as an adjuvant, reduces fecal shedding of rotavirus antigen by >95% after murine rotavirus challenge [41]. This method avoided the co-expression of interleukins by stimulating CD4 + T cells to produce inflammatory factors. The deletion of some genes of ILTV can reduce virus shedding, such as ∆UL50G, ∆UL50, ∆UL47, and UL50R42. These methods provide ideas for an ILTV vaccine to reduce virus shedding.

## 5. Conclusions

In summary, rFPV-gB significantly elevated immune protection and mitigated clinical signs, compared to the crFPV vaccine. The data indicated that rFPV-gB may be a potential vaccine against predominantly epidemic ILTV in chickens.

## Figures and Tables

**Figure 1 vaccines-08-00623-f001:**
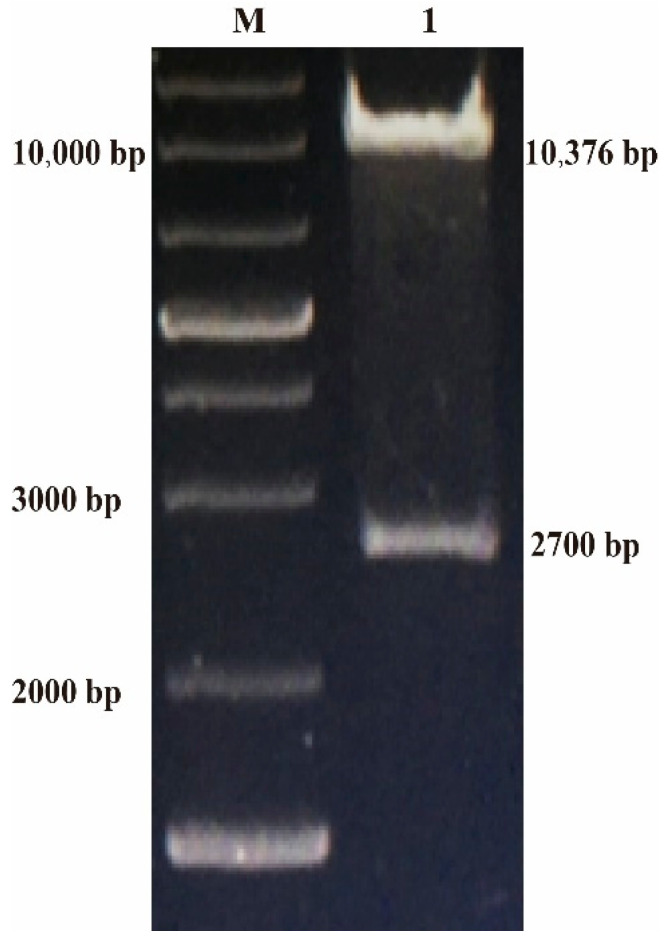
Identification of p12LS-gB by digestion with *Bam*H I and *Sma* I. After digestion with *Bam*HI and *Xho*I, the mutated synonymous gB gene of ILTV was inserted into plasmid, p12LS, which was digested with the same restriction enzyme, to form the transferring vector, p12LS-gB. The positive transferring vector plasmid p12LS-gB was identified by digestion with *Bam*HI and *Xho*I. M: 1 kb plus DNA ladder, 1: p12LS-gB digested with *Bam*H I and *Sma* I.

**Figure 2 vaccines-08-00623-f002:**
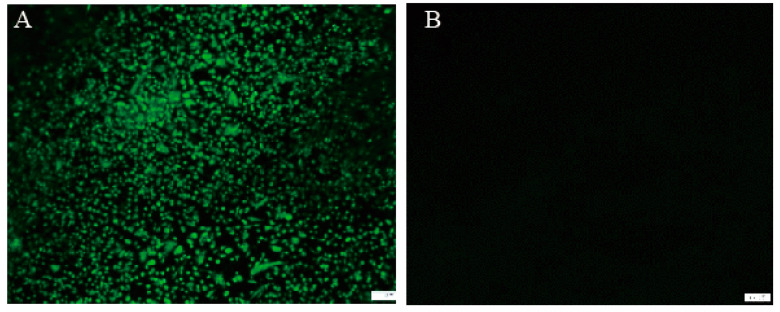
Expression of the gB gene in rFPV by IFA (×100 µm). CEF cells were infected with rFPV-gB or wt-FPV at MOI of 0.02. Expression of the ILTV gB gene was confirmed at 48 h post-infection with the chicken polyclonal antibody against the gB gene of ILTV by IFA. (**A**) rFPV-gB, (**B**) wt-FPV.

**Figure 3 vaccines-08-00623-f003:**
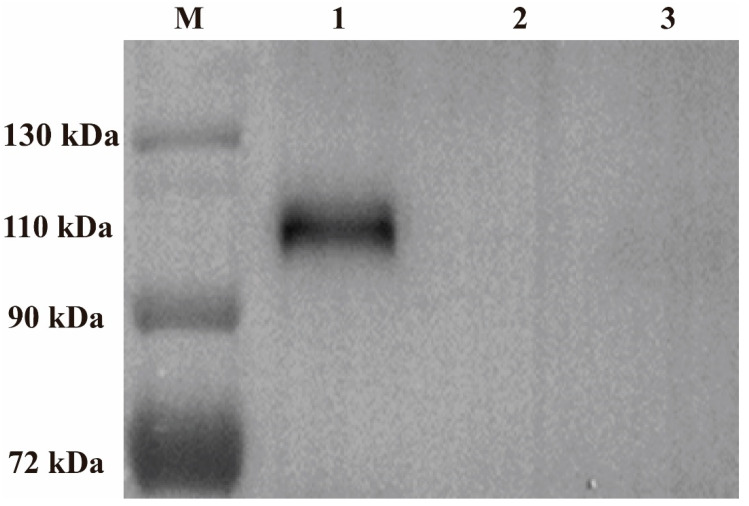
Expression of the gB gene in rFPV by Western blot. CEF cells were infected with rFPV-gB or wt-FPV at MOI of 0.1. The infected cells were harvested at 48 h post-infection, and then mixed with the sodium dodecyl sulfate (SDS) sample buffer and boiled for 10 min. Expression of ILTV gB was determined with the chicken polyclonal antibody against the gB gene of ILTV by Western blot. M: Marker, 1: rFPV-gB, 2: wt-FPV, 3: Mock CEF cells.

**Figure 4 vaccines-08-00623-f004:**
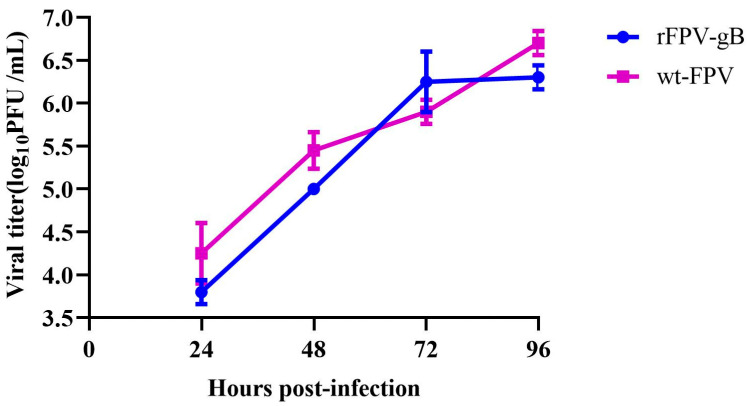
Growth kinetics of recombinants in CEF cells. CEF cells were infected with rFPV-gB or wt-FPV at a multiple of infection (MOI) of 0.02. After 1 h of fowlpox virus adsorption, cells were washed twice with PBS to remove unbound virus particles and DMEM medium containing 1% FBS was added. The infected cells were harvested by three freeze/thaw cycles at different time-points and PFUs of FPVs were determined.

**Figure 5 vaccines-08-00623-f005:**
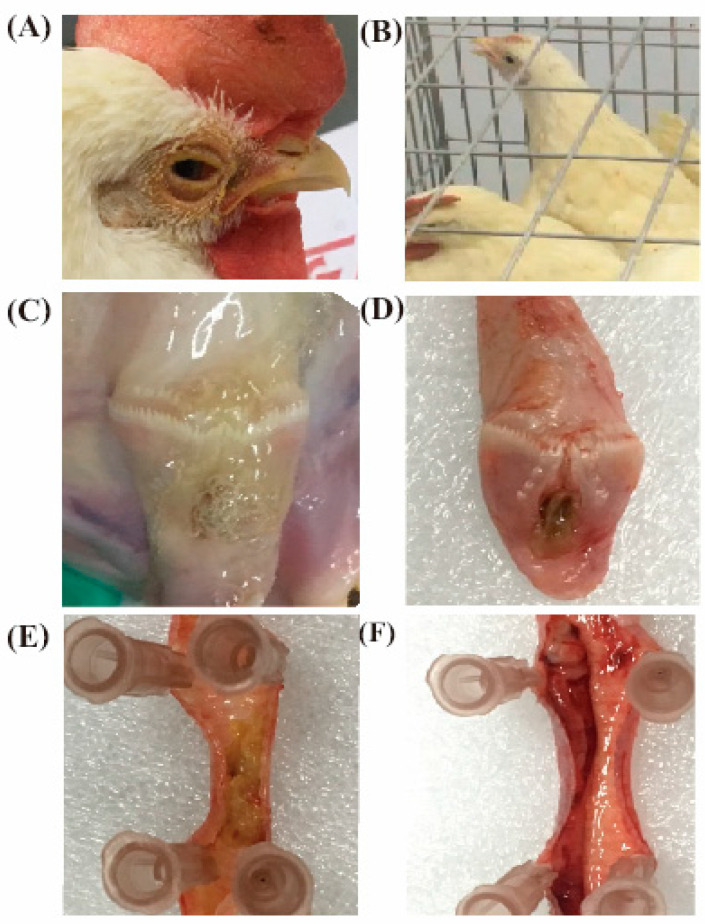
Clinical symptoms and throat tracheal lesions of chickens after challenge. (**A**) Serious conjunctivitis, (**B**) dyspnea, (**C**) mucus in the mouth and throat, (**D**–**F**) larynx and trachea with embolism of inflammatory exudate, bloody exudate, and hemorrhagic points.

**Figure 6 vaccines-08-00623-f006:**
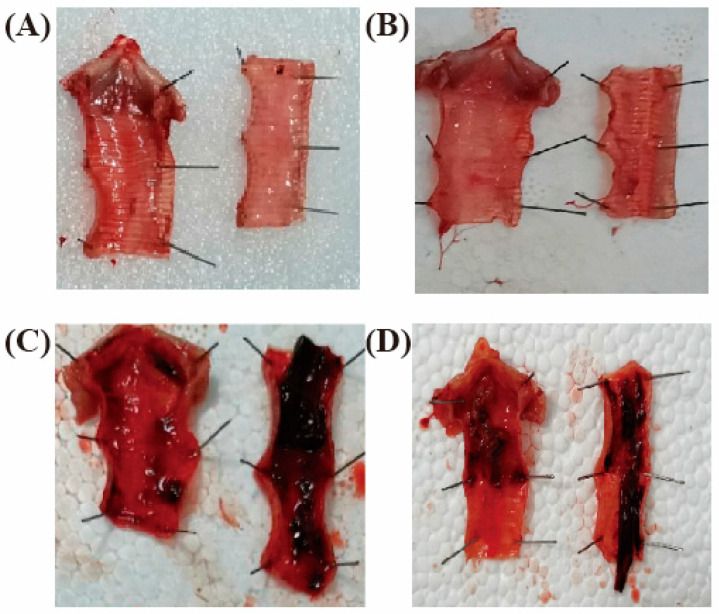
The pathological changes of inoculated chickens and control chickens after challenge. (**A**) rFPV-gB, (**B**) mild respiratory cases from the crFPV, (**C**) control group, (**D**) acute respiratory cases from the crFPV. (**A**,**B**) Larynx and trachea with congestion and bleeding, (**C**,**D**) larynx and trachea with embolism of bloody exudate.

**Figure 7 vaccines-08-00623-f007:**
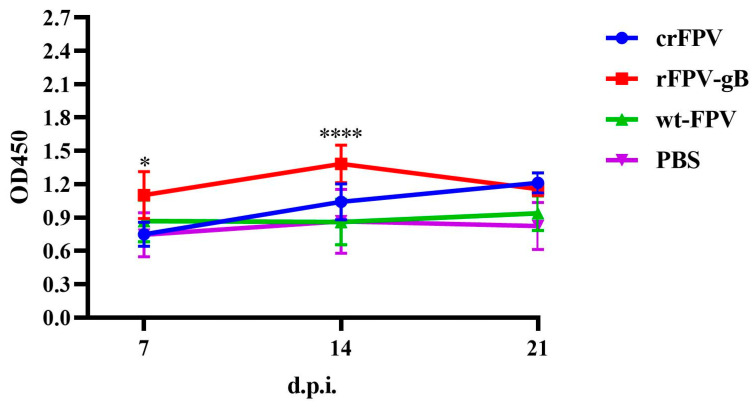
Antibody responses to ILTV after vaccination. The antibody titers to gB induced by rFPV-gB or crFPV were detected by enzyme-linked immunosorbent assay. The 96-well plates were coated with purified recombinant gB protein (0.125 µg/well) and blocked with 3% skim milk, then washed using phosphate buffered saline with tween 20 (PBST). The serum samples from immunized chickens were collected at 7, 14, and 21 days after vaccination and serially diluted 2-fold in blocking buffer in a range of 10^−1^–10^−8^, added to the wells (100 µL). The plates were then washed. Horseradish peroxidase (HRP)-labeled goat anti-chicken immunoglobulin G (IgG) was diluted (1:5000) with PBST buffer and added into plates for incubation. After washing five times, soluble tetramethyl benzidine (TMB) solution was added (100 µL), and plates were incubated in the dark at room temperature for 30 min. The reaction was terminated with 2M H_2_SO_4_ and absorbance of each well was read at a wavelength of 450 nm by a microplate absorbance reader. The antibody titers of rFPV-gB or crFPV were analyzed by graphpad 2-way analysis of variance (ANOVA), * = *p* < 0.05, **** = *p* < 0.0001.

**Table 1 vaccines-08-00623-t001:** Sequence analysis of epidemic strain I6E1 and commercial vaccine strain WG.

Class	TK (%)		ICP4		ORFB-TK		gB	
	WG	I6E1	WG	I6E1	WG	I6E1	WG	I6E1
1	100	100	99.6	99.6	99.9	99.7	99.7	99.8
2	99.5	99.5	99.7	99.7	99.7	99.6	99.7	99.8
3	99.5	99.5	99.6	99.6	99.7	99.6	99.7	99.8
4	100	100	100	100	100	99.7	100	100
5	100	100	99.6	99.6	99.9	99.7	99.7	99.8
6	99.5	99.5	99.7	99.7	99.7	99.6	99.7	99.8
7	99.6	99.6	99.9	99.8	99.7	99.7	99.9	99.9
8	100	100	100	100	99.7	100	100	100
9	100	100	99.9	99.9	99.9	99.7	99.9	99.9
10	100	100	99.6	99.6	99.7	99.5	99.7	99.6

**Table 2 vaccines-08-00623-t002:** The clinical index, morbidity, and mortality of I1E1, I6E1, WG, and I19.

Group ^a^	Dose	Challenge Route	Clinical Index (Total Score/Total Chickens) ^b^	Morbidity (Number of Cases/Total Chickens) ^c^	Mortality (Number of Deaths/Total Chickens) ^c^
I1E1	10^5^ EID_50_/per	intraocular-nasal route	0.75 (3/4)	75% (3/4)	0 (0/4)
I6E1	10^5^ EID_50_/per	intraocular-nasal route	0.75 (3/4)	75% (3/4)	0 (0/4)
WG	10^5^ EID_50_/per	intraocular-nasal route	1.5 (6/4)	100% (4/4)	0 (0/4)
	10^6^ EID_50_/per	oropharyngeal route	3 (12/4)	100% (4/4)	100% (4/4)
I19	10^5^ EID_50_/per	intraocular-nasal route	1.5 (6/4)	100% (4/4)	0 (0/4)
	10^6^ EID_50_/per	oropharyngeal route	2.5 (10/4)	100% (4/4)	50% (2/4)

^a^ Chickens in different groups challenged with 10^5^ EID_50_ or 10^6^ EID_50_ I1E1, I6E1, WG, and I19, respectively. ^b^ Total score = Clinical index/Total chickens. ^c^ Morbidity and mortality = Positive chickens /Tested chickens.

**Table 3 vaccines-08-00623-t003:** The clinical index, morbidity, and mortality of rFPVs after challenge.

Immunization Group ^a.^	Clinical Index (Total Score/Total Chickens) ^b^	Morbidity (Number of Cases/Total Chickens) ^c^	Mortality (Number of Deaths/Total Chickens) ^c^
Commercial rFPV vaccine	0.6 (6/10)	40% (4/10) ^a^	10% (1/10)
rFPV-gB	0.3 (3/10)	30% (3/10) ^a^	0 (0/10)
wt-FPV	2.5 (25/10)	100% (10/10) ^b^	60% (6/10)
PBS	2.4 (24/10)	100% (10/10)	60% (6/10)

^a^ = *p* < 0.05; ^b^ = *p* > 0.05 compared to the morbidity of the wt-FPV, all data were analyzed by Fisher’s exact test. ^c^ Morbidity or Mortality = Positive chickens/Total chickens.

**Table 4 vaccines-08-00623-t004:** Protective efficacy induced by rFPVs in SPF chickens.

Immunization Group	Immunization Dose	Mortality (Number of Deaths/Total Chickens)	Protection Index (PI)
Commercial rFPV vaccine	10^5^ PFU ^c^/0.2 mL	1/10 (10%) ^b^	83.3
rFPV-gB	10^5^ PFU/0.2 mL	0/10 (0) ^a^	100
wt-FPV	10^5^ PFU/0.2 mL	6/10 (60%) ^b^	0
PBS	0.2 mL	6/10 (60%)	0

^a^*p* < 0.05; ^b^
*p* > 0.05 compared to the mortality of the wt-FPV, ^c^. plaque forming unit (PFU), all data were analyzed by Fisher’s exact test.

**Table 5 vaccines-08-00623-t005:** Virus shedding in vaccinated chickens challenged with ILTV I19 strain.

Immunization Group	I19 Virus Isolation Rate (Positive Number/Total Number)		
	3 d.p.i. ^a^	5 d.p.i.	7 d.p.i.
Commercial rFPV vaccine	90% ^b^ (9/10) ^c^	100% (9/9)	100% (9/9)
rFPV_282E4_-gB	70% (7/10)	100% (10/10)	100% (10/10)
wt-FPV_282E4_	100% (10/10)	100% (4/4)	100% (4/4)
PBS	100% (10/10)	100% (7/7)	100% (4/4)

^a^ d.p.i. = days post-infection; ^b^ virus isolation rate = number of chickens with positive laryngeal virus isolation/total number of chicken; ^c^ positive numbers/test numbers.

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
