# Peer review of "Recombinant Fowlpox Virus Expressing gB Gene from Predominantly Epidemic Infectious Larygnotracheitis Virus Strain Demonstrates Better Immune Protection in SPF Chickens"

_vaccines, 2020, doi:10.3390/vaccines8040623_

Round 1

Reviewer 1 Report

The Authors have investigated an interesting topic and the theme has been properly described.

I would like to congratulate Authors for the good-quality of the article, the literature reported used to write the paper, and for the clear and appropriate structure. The manuscript is well written, presented and discussed, and understandable to a specialist readership.

In general, the organization and the structure of the article are satisfactory and in agreement with the journal instructions for authors. The subject is adequate with the overall journal scope.

The work shows a conscientious study in which a very exhaustive discussion of the literature available has been carried out. The introduction provides sufficient background, and the other sections include results clearly presented and analyzed exhaustively.

So, I recommend the acceptance of the paper in Vaccines.

Author Response

Response:Thank you very much for your positive evaluation of our work. Here we sincerely wish you good health and success in your work.

Reviewer 2 Report

The present study aims to describe a new vaccine against GaHV-1 or ILTV (causative agent of infectious laryngotracheitis) to better control this important respiratory disease of birds. The strategy presented (already used by a commercial vaccine) is based on a vector virus (the fowlpox virus or FPV) carrying the gene encoding the B glycoprotein (gB) of GaHV-1 (or ILTV).
Comment 1 : The difference between the gB used in this study and that of the commercial vaccine deserves to be better explained, and to say how it matches the gB of circulating strains ; to do this, the table of additional material should be included directly in the article
Comment 2 : In FIG. 1, it should be specified that the size of the bands obtained after digestion correspond well to those expected.
Comment 3 : Finally, it would be advisable to better specify the origin of the 4 wild strains used in this study and to say how they are in adequacy with the current circulating strains.
Question : The I6E1 strain is identified as IE61 in lines 365 & 366; check this difference.

Author Response

Point 1: The difference between the gB used in this study and that of the commercial vaccine deserves to be better explained, and to say how it matches the gB of circulating strains; to do this, the table of additional material should be included directly in the article.

Response1: Thanks for your kind advices. As your suggestion, we have revised in the article (line81-84; line168-172)

Point 2: In FIG. 1, it should be specified that the size of the bands obtained after digestion correspond well to those expected.
Response2:Thanks for your question, according to your suggestion, We have revised in the article (line176) and marked the size of the bands in FIG.1.

Point 3: Finally, it would be advisable to better specify the origin of the 4 wild strains used in this study and to say how they are in adequacy with the current circulating strains.

Response3: Thanks for your kind advices. We have revised in the article (line66-68).

Question: The I6E1 strain is identified as I6E1 in lines 365 & 366; check this difference.

Response4: Thank you very much for correcting the writing error for me. We have revised in the article (line345).

This manuscript is a resubmission of an earlier submission. The following is a list of the peer review reports and author responses from that submission.